# Background Factors Affecting Visual Acuity at Initial Visit in Eyes with Central Retinal Vein Occlusion: Multicenter Study in Japan

**DOI:** 10.3390/jcm10235619

**Published:** 2021-11-29

**Authors:** Mineo Kondo, Hidetaka Noma, Masahiko Shimura, Masahiko Sugimoto, Yoshitsugu Matsui, Kumiko Kato, Yoshitsugu Saishin, Masahito Ohji, Hiroto Ishikawa, Fumi Gomi, Kensaku Iwata, Shigeo Yoshida, Sentaro Kusuhara, Hiromasa Hirai, Nahoko Ogata, Takao Hirano, Toshinori Murata, Kotaro Tsuboi, Motohiro Kamei, Takamasa Kinoshita, Soichiro Kuwayama, Yoshio Hirano, Manami Ohta, Kazuhiro Kimura, Kei Takayama, Masaru Takeuchi, Yoshihiro Takamura, Fumiki Okamoto, Yoshinori Mitamura, Hiroto Terasaki, Taiji Sakamoto, on behalf of Japan Clinical Retina Study (J-CREST) Group

**Affiliations:** 1Department of Ophthalmology, Mie University Graduate School of Medicine, Tsu 514-8507, Japan; sugmochi92@gmail.com (M.S.); footboyslim366@gmail.com (Y.M.); k-kato@med.mie-u.ac.jp (K.K.); 2Department of Ophthalmology, Tokyo Medical University Hachioji Medical Center, Hachioji 193-0998, Japan; noma-hide@umin.ac.jp (H.N.); masahiko@v101.vaio.ne.jp (M.S.); 3Department of Ophthalmology, Shiga University of Medical Science, Otsu 520-2192, Japan; saishin@belle.shiga-med.ac.jp (Y.S.); eye.ohji@gmail.com (M.O.); 4Department of Ophthalmology, Hyogo College of Medicine, Nishinomiya 663-8501, Japan; ohmyeye@gmail.com (H.I.); fgomi@hyo-med.ac.jp (F.G.); 5Department of Ophthalmology, Kurume University School of Medicine, Kurume 830-0011, Japan; iwata_kensaku@kurume-u.ac.jp (K.I.); usyosi@gmail.com (S.Y.); 6Division of Ophthalmology, Department of Surgery, Kobe University Graduate School of Medicine, Kobe 650-0017, Japan; kusu@med.kobe-u.ac.jp; 7Department of Ophthalmology, Nara Medical University, Kashihara 634-8521, Japan; hirai-masa@naramed-u.ac.jp (H.H.); ogata@naramed-u.ac.jp (N.O.); 8Department of Ophthalmology, Shinshu University School of Medicine, Matsumoto 390-8621, Japan; takaoh@shinshu-u.ac.jp (T.H.); murata@shinshu-u.ac.jp (T.M.); 9Department of Ophthalmology, Aichi Medical University, Nagakute 480-1195, Japan; kotarotsuboi@gmail.com (K.T.); motokamei@gmail.com (M.K.); 10Department of Ophthalmology, Sapporo City General Hospital, Sapporo 060-8604, Japan; knst129@gmail.com; 11Department of Ophthalmology & Visual Science, Nagoya City University Graduate School of Medical Sciences, Nagoya 467-8601, Japan; gifu8847@yahoo.co.jp (S.K.); yoshio.hirano@gmail.com (Y.H.); 12Department of Ophthalmology, Yamaguchi University Graduate School of Medicine, Ube 755-8505, Japan; mohta@yamaguchi-u.ac.jp (M.O.); k.kimura@yamaguchi-u.ac.jp (K.K.); 13Department of Ophthalmology, National Defense Medical College, Tokorozawa 359-8513, Japan; keitaka1234@hotmail.com (K.T.); masatake@ndmc.ac.jp (M.T.); 14Department of Ophthalmology, Faculty of Medical Sciences, University of Fukui, Yoshida 910-1193, Japan; ytakamura@hotmail.com; 15Department of Ophthalmology, Faculty of Medicine, University of Tsukuba, Tsukuba 305-8575, Japan; ultrasoundbiomicroscopy@mail.goo.ne.jp; 16Department of Ophthalmology, Institute of Biomedical Sciences, Tokushima University Graduate School, Tokushima 770-8503, Japan; ymitaymitaymita@yahoo.co.jp; 17Department of Ophthalmology, Kagoshima University Graduate School of Medical and Dental Sciences, Kagoshima 890-8544, Japan; hirototerasaki112@gmail.com (H.T.); tsakamot@m3.kufm.kagoshima-u.ac.jp (T.S.)

**Keywords:** central retinal vein occlusion, visual acuity, multicenter study

## Abstract

Purpose: To determine the baseline characteristics of patients with central retinal vein occlusion (CRVO) that were significantly associated with the best-corrected visual acuity (BCVA) at the initial examination. Methods: This was a retrospective multicenter study using the medical records registered in 17 ophthalmological institutions in Japan. Patients with untreated CRVO (≥20-years-of-age) who were initially examined between January 2013 and December 2017 were studied. The patients’ baseline factors that were significantly associated with the BCVA at the initial examination were determined by univariate and multivariate linear regression analyses. Results: Data from 517 eyes of 517 patients were analyzed. Univariate analyses showed that an older age (*r* = 0.194, *p* < 0.001) and the right eye (*r* = −0.103, *p* < 0.019) were significantly associated with poorer BCVA at the initial visit. Multivariate analyses also showed that an older age (*β* = 0.191, *p* < 0.001) and the right eye (*β* = −0.089, *p* = 0.041) were significantly associated with poorer BCVA at the initial visit. Conclusions: The results indicate that an older age, a known strong factor, and the right eye were significantly associated with poorer BCVA at the initial visit to the hospital. These results suggest that functional and/or anatomical differences between the right and left eyes may be involved in these results.

## 1. Introduction

A central retinal vein occlusion (CRVO) is a vascular disorder of the retina that can result in a severe reduction of vision. The exact pathogenesis of CRVO has not been definitively determined, but several factors, including compression by a sclerotic central retinal artery, degenerative changes of the vessel wall, hemodynamic disturbances, blood hypercoagulability, and partial thrombosis, may be involved [1,2,3,4]. The global prevalence of CRVO is estimated to be about 0.08–0.13%, with a higher incidence in older individuals [5,6,7]. The common causes of vision reduction in CRVO are macular edema, vitreous hemorrhage due to retinal neovascularization, and rubeotic glaucoma due to iris/angle neovascularization.

At present, several different types of treatments have been used for eyes with a CRVO, including laser photocoagulation [8], intravitreal injection of corticosteroids including triamcinolone acetonide [9], intravitreal implantation of dexamethasone [10], and intravitreal injections of anti-vascular endothelial growth factor (anti-VEGF) agents [11,12]. However, the visual prognosis of CRVO is still not good [13,14,15].

It is known that the baseline best-corrected visual acuity (BCVA) is the strongest predictor for the final BCVA in eyes with CRVO [16,17,18,19,20]. Therefore, the factors involved in visual acuity at the baseline can be indirectly related to the final visual outcome. However, a PubMed search did not extract any study examining the baseline factors of patients that were significantly associated with poorer BCVAs at the initial visit to the hospital.

Thus, the purpose of this study was to determine the baseline characteristics of patients with a CRVO that were significantly associated with the baseline BCVA. To accomplish this, we conducted a multicenter collaborative study of 517 eyes of 517 patients with CRVO.

## 2. Materials and Methods

### 2.1. Study Design and Approval

This was a multicenter retrospective study analyzing the medical records of 31 retina specialists at 17 ophthalmological institutions in Japan. The study protocol was approved by the Ethics Committees of all participating centers, with the main Ethics Committee at the Mie University Hospital (#H2018-034). A written informed consent was not obtained from the subjects because of the retrospective nature of this study. Instead, a home page was created with information on the purpose of this study for the subjects to read. We emphasized that any subject could opt out of the study at any time by telephone, fax, or e-mail. The need of waived informed consent was approved by the Ethics Committee at the Mie University Hospital (#H2018-034). The study was also registered on the International Clinical Trial Registry Platform (UMIN Clinical Trials Registry, R000037123, http://www.umin.ac.jp/ctr/index-j.htm) (accessed on 10 May 2018). The data acquired for the analysis were anonymized before they were examined. The procedures used conformed to the tenets of the Declaration of Helsinki of the World Medical Association. All procedures were carried out in accordance with “Ethical Guidelines for Medical Research Involving Human Subjects” in Japan.

### 2.2. Subjects

Treatment-naïve CRVO patients who visited the participating hospitals between January 2013 and December 2017, and were followed for more than 12 months were enrolled. This inclusion criterion was made because this study was originally designed to investigate the treatments and prognosis of CRVO over a 12-month period. Only CRVO patients whose age was ≥20 years, and whose interval between the onset of symptoms to the initial visit to the hospital was ≤12 month were included. Eyes with hemi-CRVO were excluded. CRVO eyes with glaucoma or advanced cataract (≥Grade 3) [21] were also excluded. If patients had CRVO in both eyes and visited the hospital between January 2013 and December 2017, only the eye with an earlier onset was studied.

The diagnosis of CRVO was made by each retinal specialist based on the presence of retinal edema, optic disc hyperemia or edema, scattered superficial and deep retinal hemorrhages, and venous dilation in the fundus photographs and optical coherence tomographic images. These criteria were based on an earlier study [5].

The age, sex, affected eye (right or left), presence of systemic hypertension and diabetes mellitus, and the interval between the onset of the symptoms to the initial visit to the hospital were collected from the medical records. Initially, we wanted to include the factors of hyperlipidemia, cardiovascular diseases, and smoking, however, we did not include these factors in this analysis because, in many cases, there was no accurate description in the medical charts whether the patients were asked whether they had these factors. The best-corrected visual acuity (BCVA) and central macular thickness (CMT) at the initial visit were also collected.

### 2.3. Best-Corrected Visual Acuity and Central Macular Thickness

The BCVA was measured with a standard Japanese decimal visual acuity chart at 5 m. The visual acuities of “counting fingers” and “hand motion” were quantified to decimal values as 0.014 and 0.004, respectively, based on a study [22]. The decimal values were converted to the logarithm of the minimal angle of resolution (logMAR) units for the statistical analyses.

The CMT was measured by spectral-domain optical coherence tomography (OCT) at the initial visit using Spectralis (Heidelberg Engineering, Heidelberg, Germany), Cirrus (Carl Zeiss Meditec, Inc., Dublin, CA, USA), Triton (Topcon, Tokyo, Japan), or RS-3000 (Nidek, Gamagori, Japan). The average thickness within a 1-mm diameter of the central macular area was used for the analyses. If the average thickness within the 1-mm diameter was not measured, the average foveal thickness of the vertical and horizontal scans was calculated and used as the value of the CMT.

### 2.4. Classification of Ischemic Status

All CRVO eyes were classified as the ischemic-type or nonischemic-type based on the fluorescein angiographic (FA) findings performed at the initial visit. The classical definition of the CVO Study [8] was used: the CRVO was classified as the ischemic-type if the eye had at least a 10-disc area of retinal capillary nonperfusion within the area of a standard photographic field. If FA had not been performed, the eye was classified as ischemic-type when at least two of the following five findings were present: (1) massive retinal hemorrhage associated with prominent tortuosity of the retinal vessel; (2) decimal BCVA < 0.1; (3) loss of the 1-2e isopter in Goldmann visual field; (4) apparent relative afferent pupillary defect; or (5) ERG b-wave amplitude reduced to ≤60% of that of the normal fellow eye [23,24,25]. This classification was done at the individual centers. When these tests were not done adequately, we accepted that the judgement of ischemia was based on the clinicians’ impression as noted in the clinical record.

### 2.5. Statistical Analyses

The probabilities of the presence of a CRVO between men and women or between right and left eyes were compared by binomial tests. Univariate and multivariate linear regression analyses were used to determine the background factors which affected the BCVA at the initial visit to the hospital. The BCVAs at the initial visit were used as the dependent variables. The independent variables were the age, sex, affected eye (right or left), presence of systemic hypertension, presence of diabetes mellitus, and interval from symptom onset to initial visit to the hospital. The coefficients of correlation (*r*) and *p*-values were calculated for the univariate linear regression analysis, and standardized partial regression coefficient (*β*) and *p*-values were calculated for the multivariate linear regression analyses for the six independent variables. To determine the background factors which were associated with a BCVA poorer than 1.0 logMAR units (0.1 decimal BCVA) at the initial visit to the hospital, the logistic regression analysis using stepwise forward selection method was performed. The same six independent valuables were used.

After confirming that the data were approximately normally distributed, the significance of the differences in the BCVA (logMAR unit) or CMT between the right and left eye were compared by unpaired *t*-tests. The results were considered statistically significant when *p* < 0.05. Analyses were performed with SPSS software (IBM SPSS Statistics 25; IBM Corp., Armonk, NY, USA).

## 3. Results

### 3.1. Demographic Information

Data of 588 eyes with a CRVO were collected from 17 ophthalmological institutions in Japan. Seventy-one eyes (12%) were excluded due to not meeting the inclusion criteria or missing data. In the end, 517 eyes were used for the analyses.

The demographic information of the 517 eyes of 517 patients with a CRVO is shown in Table 1. The mean ± SD age of the patients was 69.9 ± 12.2 years with a range of 22 to 94 years. Fifty-eight percent of the patients were ≥70 years (Figure 1A). The mean age was significantly higher in women (71.9 ± 12.1 years) than men (68.4 ± 12.1 years, *p* = 0.001, Figure 1B). Only 10 (1.9%) eyes were under 40-years-of-age.

Of the 517 patients with a CRVO, 296 were men (57.3%) and 221 were women (42.7%). The incidence of CRVO was significantly higher in men than in women (*p* = 0.001, binominal test, Table 1). Systemic hypertension was present in 334 patients (64.6%), and 87 patients (16.8%) had diabetes mellitus. The mean interval between the onset of symptoms to the initial visit to hospital was 6.4 ± 6.9 weeks (range, 0–51 weeks). We also noted that 467 of 517 patients (90.3%) visited a hospital within 12 weeks of the onset of the symptoms.

The mean ± SD of the BCVA was 0.72 ± 0.55 logMAR units (0.19 decimal BCVA) with a range of −0.18 to 2.30 logMAR units. The number of eyes with a BCVA worse than 0.3 logMAR units (0.50 decimal BCVA) was 355 (88.7%), and the number of eyes with a BCVA worse than 1.0 logMAR units (0.1 decimal BCVA) was 113 (21.9%). The mean ± SD central macular thickness (CMT) at the initial visit was 632 ± 237 μm with a range of 62 to 1456 μm. A CMT of 62 μm was found in one patient with ischemic CRVO who visited a hospital 11 months after the onset of the symptoms, and the macula was severely atrophic.

At the initial visit, 122 eyes (23.6%) were classified as the ischemic-type, 377 eyes (72.9%) were the nonischemic-type, and 18 eyes (3.5%) were unclassifiable.

### 3.2. Background Factors Affecting Visual Acuity at Initial Visit

To determine the patients’ background factors that affected the BCVA at the initial visit to the hospital, we performed univariate and multivariate linear regression analyses (Table 2). Univariate linear regression analyses showed that the age (*r* = 0.194, *p* < 0.001) and the affected eye (right or left, *r* = −0.103, *p* = 0.019) were significantly associated with the BCVA at the initial visit to the hospital (Table 2, left panel). Multivariate linear regression analyses also identified the same two independent factors: the age (*β* = 0.191, *p* < 0.001) and the affected eye (*β* = −0.089, *p* = 0.041) as independent factors which affected the BCVA at the initial visit (Table 2, right panel).

We also sought to identify the background factors that were significantly related to a BCVA worse than 1.0 logMAR units (0.1 decimal BCVA) at the initial visit to the hospital using logistic regression analyses (Table 3). The same six dependent values were used for the independent variables. The results showed that the same two factors, an older age (*β* = 0.243, *p* = 0.016) and the right eye (*β* = −0.195, *p* = 0.038), were significantly associated with a BCVA worse than 1.0 logMAR units at the initial visit.

### 3.3. Relationship between Age and BCVA at Initial Visit to Hospital

The BCVAs at the initial visit to the hospital are plotted against the age in Figure 2A. A linear regression fit to the data indicated that a 10-year increase of age was accompanied by an approximately −0.0876 logMAR unit change of the BCVA at the initial visit to the hospital (red line, Figure 2A). A plot of the BCVA at the initial visit for each age group showed that the mean BCVA was 0.50 logMAR units (0.32 decimal BCVA) for the eyes <50-years-of-age. The mean BCVA became gradually worse with age increasing, and reached 0.83 logMAR units (0.15 decimal BCVA) for the eyes of patients ≥80 years (Figure 2B).

### 3.4. Comparisons of Visual Acuity and Central Macular Thickness for Right and Left Eyes

Because both linear and logistic regression analyses demonstrated that the affected eye (right or left) was significantly associated with the BCVA at the initial visit to the hospital, we next plotted the BCVA of the right and left eyes in all 517 CRVO patients (Figure 3A). The means ± SDs of the BCVA (logMAR units) at the initial visit to the hospital was significantly better in the left eye (0.66 ± 0.53 logMAR units) than in the right eye (0.78 ± 0.56, logMAR units; *p* = 0.019). We also found that the percentage of eyes with BCVA worse than 1.0 logMAR units (0.1 decimal BCVA) was 30.0% (72/240 eyes) in the right eye, and it was 22.7% in the left eye (63/277 eyes).

In our CRVO patients, we had 11 patients who had CRVO in both eyes during the observation period of 12 months. We compared the BCVA of the right and left eyes in all CRVO eyes, including both eye data of these 11 patients. We confirmed that the difference in visual acuity between the right and left eyes was still statistically significant (*p* = 0.020).

We also compared the CMT of the right to that of the left eye at the initial visit, and found that the mean CMT was significantly thicker in the right eye (655 ± 253 μm) than in the left eye (613 ± 218 μm; *p* = 0.011; Figure 3B). There was also a significant correlation between the BCVA and CMT at the initial examination (*r* = 0.200, *p* < 0.001, Figure 4).

The percentage of eyes with ischemic CRVO was also compared between the right and left eyes. The percentage of ischemic CRVO tended to be slightly higher in the right eye (25.8%, 62/240 eyes) than in the left eye (21.7%, 60/277 eyes), but this difference was not statistically significant (*p* = 0.319, chi-squared test).

We also sought to determine whether there was any difference in the interval from the symptom onset to the initial visit to the hospital between the right eye group and the left eye group. The mean interval from symptom onset to initial visit to the hospital was slightly longer in the right eye (6.8 ± 7.0 weeks) than in the left eye (6.1 ± 6.9 weeks), but this difference was not statistically significant (*p* = 0.251).

## 4. Discussion

In this multicenter study, we examined the baseline factors of patients with CRVO in Japan, and sought to determine the factors that were significantly associated with the BCVA at the initial examination. The baseline factors of the patients with CRVO were similar to recent real-world data of CRVO: the average age of the patients was about 70 years, and the patients had a higher prevalence of hypertension [13,15,19,26,27].

In our CRVO cohort, there were significantly more men (57.3%) than women (42.7%, *p* = 0.001, Table 1). Another recent study of CRVO in Japan also showed that the prevalence was higher in men (64.7%) than women (35.3%) [19], whereas global epidemiological studies have shown that the prevalence of CRVO did not differ significantly between the sexes [5,7]. Real-world reports of CRVO in the USA [15], UK [26], and Germany [13] also reported no significant difference in the prevalence of CRVO between men and women, although recent results of the IRIS registry reported that the prevalence of CRVO was slightly, but significantly, higher in men (50.4%) than women (49.6%) [28]. We assume that the higher prevalence of CRVO in men in Japan may be related to the higher prevalence of hypertension and smoking in men in Japan [29,30]. Analyses of data from the National Surveys of Japan have shown that there was a clear trend for a decrease in the prevalence of hypertension in women, but not in men, over the past 30 years in Japan [31].

We also found that an older age was associated with poorer BCVA at the initial examination. This was not too surprising because there have been many past studies reporting that an older age was associated with poorer final visual outcomes in eyes with CRVO [16,18,19,20]. We assume that the age-related sclerotic changes in the arteries, and the degenerative changes of the vessel walls may be associated with this finding. On the other hand, we cannot rule out the possibility that age-related mild cataract might have affected our results because we excluded CRVO patients with severe cataracts (≥grade 3).

The most unexpected findings was that the difference between the right and left eyes affected the baseline BCVA at initial examination in eyes with CRVO. Both univariate and multivariate linear regression analyses demonstrated that the presence of the CRVO in the right eye was significantly associated with poorer BCVA at the initial visit (Table 2 and Figure 3A). We also found that the CRVO in the right eye was significantly related to a BCVA poorer than 1.0 logMAR (0.1 decimal BCVA; Table 3). This difference in the BCVA between the right and left eyes appeared to be related to the degree of macular edema because the CMT was also significantly greater in the right eye than in the left eye (Figure 3B), and there was a significant correlation between the visual acuity and the CMT (Figure 4).

Our PubMed search showed that our report is the first to report of a significant difference in the BCVA between the right and the left eyes in eyes with CRVO. The exact reason for the difference in the initial BCVA between the two eyes in CRVO was not determined. We initially suspected that the right eyes had a higher frequency of the ischemic-type of CRVO. Although, the ratios of ischemic CRVO tended to be slightly higher in the right eye (25.8%) than in the left eye (21.7%), this difference was not statistically significant (*p* = 0.319).

We need to consider the possibility that the poor initial BCVA in the right eye might be just a simple coincidence due to the small sample size. However, the results of the recent IRIS Registry demonstrated that there was a significant laterality between the right and left eyes in CRVO [28]. For example, the incidence of CRVO was not the same between the right and left eye, but was significantly higher in the left eye (49.1% vs. 50.9%). Although the IRIS Registry did not show any data of the visual acuity, this result is interesting because it suggests the possibility that there may be some left–right differences in retinal vascular disorders.

The difference in the severity of the CRVO between the right and left eyes may be due to some functional and/or anatomical differences between the right and left eyes. For example, it is known that the blood pressure measured on the right arm is slightly, but significantly, higher than that on the left arm [32,33]. Some investigators have suggested that the blood pressure might be higher in the right brachial artery compared to the left because the brachiocephalic artery is located nearer to the source of the pressure, and is aligned more in the direction of the blood flow of the ascending aorta [34]. These slight differences in blood pressure between the right and left circulations might cause different degrees of compressions by the central retinal artery. It is also known that the left common carotid artery (CCA) stems directly from the arch of aorta, and is more affected by aortic arch pressure (hydrostatic pressure), whereas the right CCA stems from the extension (innominate artery) of the ascending aorta, and is subjected to pressure from the ascending aortic blood flow (dynamic pressure). These anatomical differences in the left and right CCA might have contributed to the difference in the visual acuity between the right and left eyes of the CRVO. At present, it is still difficult to find a satisfactory explanation for the laterality of the initial visual acuity in CRVO.

There are four major limitations in this study. The first limitation is the retrospective nature of this study. We analyzed the factors affecting the baseline BCVA using only six background factors of the patients because we were able to extract only these findings from all the medical records. Other factors, such as smoking, hyperlipidemia, and arterial thromboembolic diseases, could not be included because there were many cases where the presence or absence of these factors was not recorded in the medical charts. If these factors had been included, a more detailed analysis would have been possible.

The second limitation is that all subjects were of the same ethnicity, i.e., Japanese. The results obtained in this study may be specific for the Japanese. Further studies are needed to verify whether the right–left difference can be a significant factor affecting the initial visual acuity of CRVO in patients of other ethnicities.

The third limitation is that the CMT was measured by different OCT devices. In this study, four OCT devices were used in our 17 institutes. A chi-squared test was performed to analyze whether a particular OCT device was significantly used more frequently for the right or left eye. The results showed that there was no significant trend that a particular OCT device was used more frequently for the right or left eye (*p* = 0.664). Therefore, we think that it is unlikely that the differences in the OCT device used in this study caused the left–right differences in the CMT.

The fourth limitation is that we did not use a uniform definition for the ischemic-type of CRVO, but used a modified definition using the combination of various clinical findings. This is because this study was a retrospective study using medical records. Initially, we tried to define the ischemic-type CRVO based on the results of the fluorescein angiography used in the CVO Study [8]. However, our preliminary results showed that the fluorescein angiography was not performed in more than one-half of our CRVO cohort. Therefore, for the CRVO eyes where fluorescein angiography was not performed, we tried to determine the ischemic-type comprehensively by combining the results of several clinical findings based on a paper by Hayreh et al. [23]. They demonstrated that the combination of several clinical findings can be used to determine the ischemic-type CRVO with a high degree of accuracy.

In summary, we have presented the baseline characteristics of patients with CRVO in Japan, and determined which of these factors were significantly associated with the BCVA at the initial examination. We found that age, which is known to affect the BCVA in eyes with CRVO, and the right and left eye differences can be other factors affecting the visual acuity at the initial visit. Although the mechanism for this difference was not determined, the results suggest that there may be a new baseline factor which can influence the visual acuities in eyes with CRVO.

## Figures and Tables

**Figure 1 jcm-10-05619-f001:**
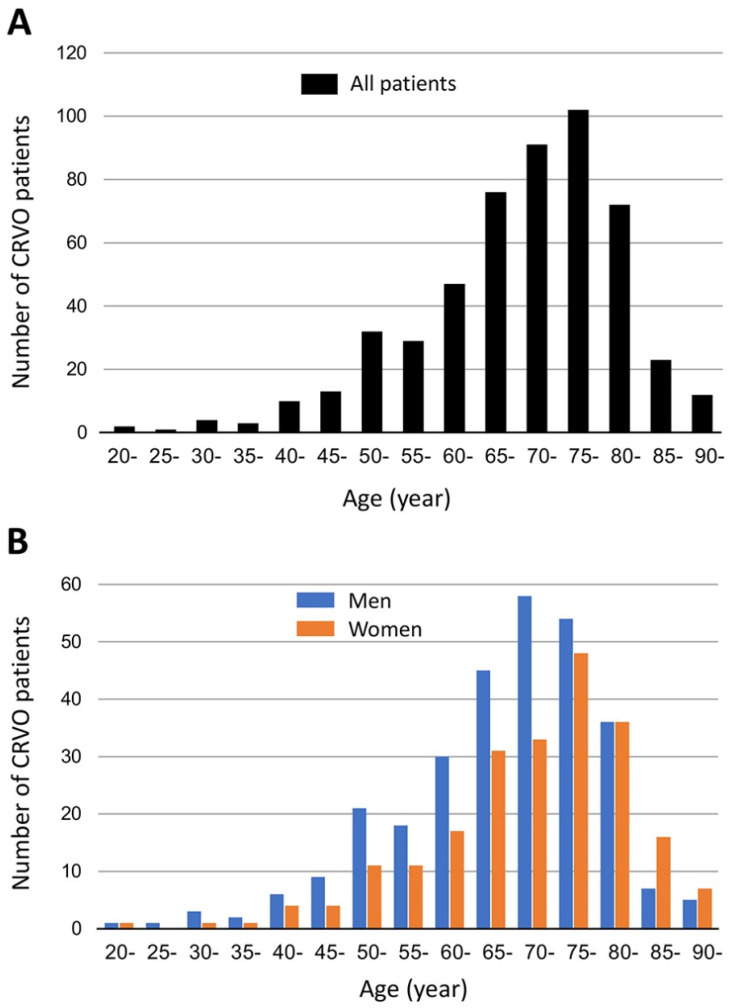
(**A**) Number of patients with central retinal vein occlusion (CRVO) for each 5-year age group. The mean ± standard deviations (SD) age of the patients was 69.9 ± 12.2 years with a range of 22 to 94 years. There were 58.0% of the CRVO patients who were 70 years or older. (**B**) Number of patients with CRVO for each age group shown by sex. The mean age of the CRVO patients was significantly higher in women (71.9 ± 12.1 years) than in men (68.4 ± 12.1 years, *p* = 0.001).

**Figure 2 jcm-10-05619-f002:**
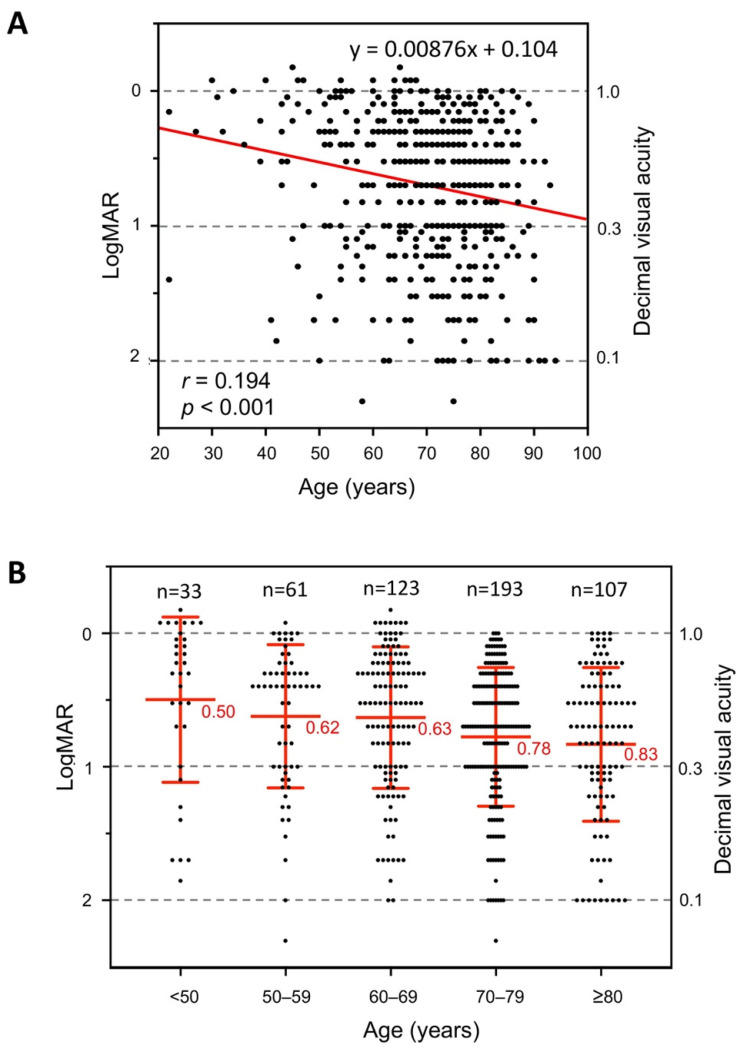
(**A**) The best-corrected visual acuity (BCVA) in logarithm of the minimal angle of resolution (logMAR) units at the initial visit to hospital plotted against the age. A linear regression fit to the data is shown by the red line *(r* = 0.194, *p* < 0.001). (**B**) The BCVA (logMAR units) at the initial visit to the hospital plotted for each age group. The mean BCVA worsened with increasing age. The red bars indicate the means ± SDs.

**Figure 3 jcm-10-05619-f003:**
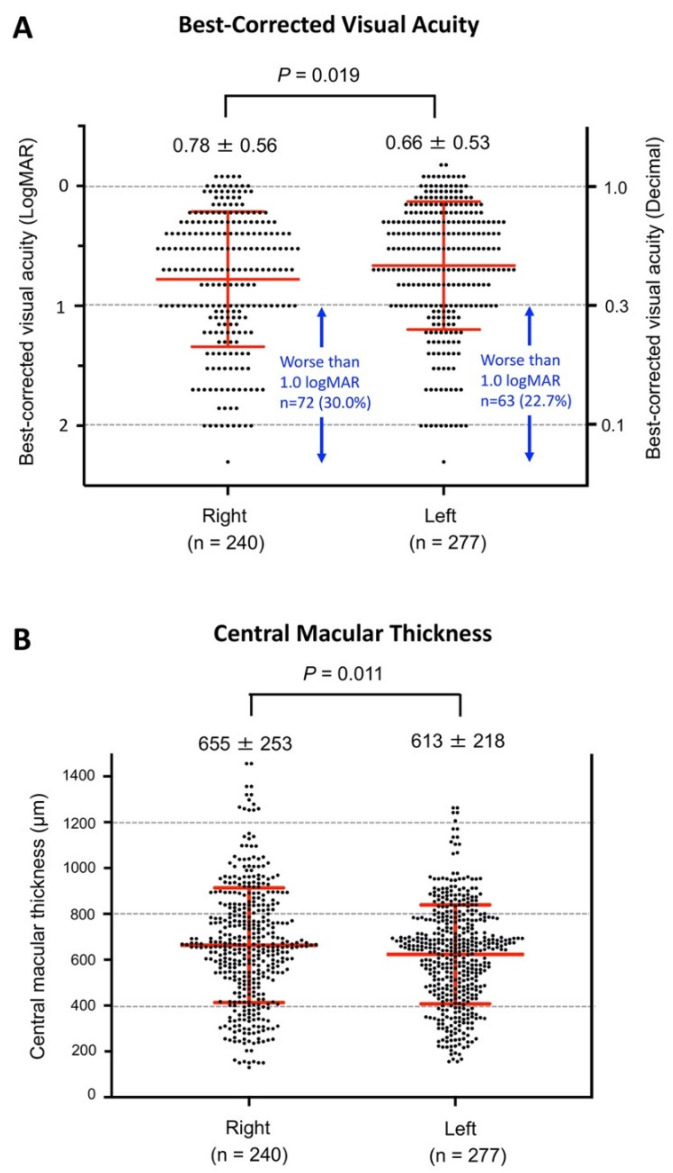
(**A**) Comparisons of the BCVA (in logMAR units) between the right and left eyes at the initial examination in all 517 CRVO patients. The mean ± SD of the BCVA (logMAR unit) was significantly better in the left eye (0.66 ± 0.53) than in the right eye (0.78 ± 0.56, *p* = 0.019). The percentage of eyes with BCVA worse than 1.0 logMAR units (0.1 decimal BCVA) was 30.0% (72/240 eyes) in the right eye, whereas it was 22.7% in the left eye (63/277 eyes). (**B**) Comparisons of central macular thickness (CMT) between the right and left eyes at the initial examination in all 517 CRVO patients. The mean CMT was significantly thicker in the right eye (655 ± 253 μm) than in the left eye (613 ± 218 μm, *p* = 0.011).

**Figure 4 jcm-10-05619-f004:**
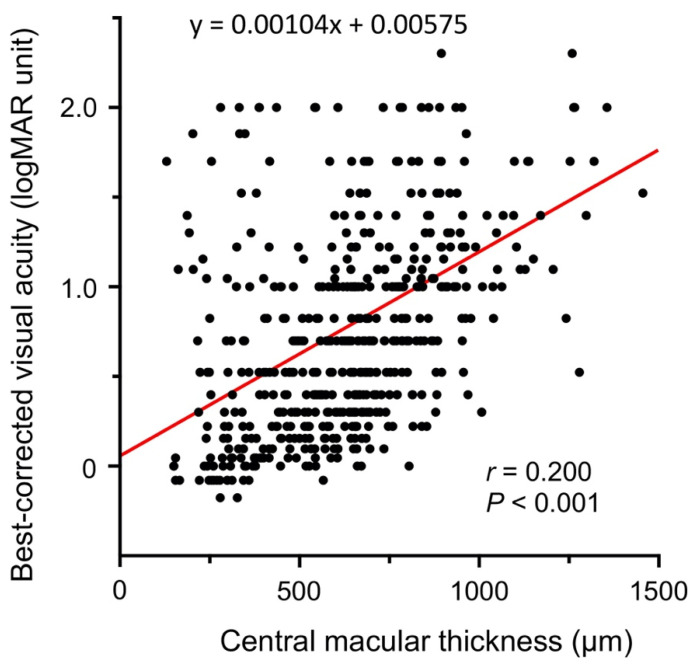
Plot of BCVA (in logMAR units) against central macular thickness (CMT) at the initial examination. There was a significant correlation between the BCVA and CMT at the initial examination (*r* = 0.200, *p* < 0.001). The best-fit linear regression line is also shown by the red line.

**Table 1 jcm-10-05619-t001:** Clinical characteristics of 517 eyes of 517 patients with CRVO at the initial visit to the hospital.

Parameter	Value	*p*-Value
Number of eyes/subjects	517/517	
Age, mean ± SD (range), years	69.9 ± 12.2 (22 to 94)	
Sex		
Men (%)	296 (57.3)	
Women (%)	221 (42.7)	0.001 *
Affected eye		
Right (%)	240 (46.4)	
Left (%)	277 (53.6)	0.111
Hypertension (%)	334 (64.6)	
Diabetes mellitus (%)	87 (16.8)	
Interval from symptom onset to initial visit to hospital, mean ± SD (range), in weeks	6.4 ± 6.9 (0 to 51)	
Best-corrected visual acuity, mean ± SD (range), logMAR	0.72 ± 0.55 (−0.18 to 2.30)	
Central macular thickness, mean ± SD (range), µm	632 ± 237 (62 to 1456)	
Ischemic status at initial visit to hospital		
Ischemic (%)	122 (23.6)	
Nonischemic (%)	377 (72.9)	
Unclassifiable (%)	18 (3.5)	

SD, standard deviation; logMAR, logarithmic minimum angle of resolution. The probabilities of the presence of a CRVO between men and women or between right and left were compared by a binomial test. * *p* < 0.05.

**Table 2 jcm-10-05619-t002:** Factors affecting best-corrected visual acuity (in logMAR units) at the initial visit to the hospital in patients with CRVO. Results of univariable and multivariate linear regression analyses are shown.

Independent Variables	Univariate Regression Analysis		Multivariate Regression Analysis	
	*r*	*p-*Value	*β*	*p-*Value
Age (years)	0.194	<0.001 **	0.191	<0.001 *
Sex (men/women)	0.005	0.916	0.023	0.586
Affected eye (right/left)	−0.103	0.019 *	−0.089	0.041 *
Hypertension	0.035	0.432	0.013	0.764
Diabetes mellitus	−0.043	0.325	−0.045	0.303
Interval from symptom onset to initial visit to hospital (weeks)	0.008	0.848	−0.008	0.849

Correlation coefficient (*r*). Standardized partial regression coefficient (*β*), and *p* value are shown for six independent variables, which can affect best-corrected visual acuity (logMAR unit) at the initial visit to the hospital in patients with CRVO. * *p* < 0.05 was considered significant. ** *p* < 0.01.

**Table 3 jcm-10-05619-t003:** Results of logistic regression analysis to identify the baseline factors that were related to BCVA worse than 1.0 logMAR units (0.1 decimal BCVA).

Independent Variables	*β*	*p-*Value
Age (years)	0.243	0.016 *
Sex (men/women)	0.031	0.741
Affected eye (right/left)	−0.195	0.038 *
Hypertension	0.002	0.984
Diabetes mellitus	−0.122	0.213
Interval from symptom onset to initial visit to hospital (weeks)	−0.144	0.155

Standardized partial regression coefficient (*β*), and *p*-values are calculated for six independent variables which are associated with visual acuities poorer than 1.0 logMAR (0.1 decimal BCVA) at the initial visit to the hospital in eyes with CRVO. * *p* < 0.05 was considered significant.

## Data Availability

All data generated or analyzed during this study are included in this published article.

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
