# Peer review of "Background Factors Affecting Visual Acuity at Initial Visit in Eyes with Central Retinal Vein Occlusion: Multicenter Study in Japan"

_jcm, 2021, doi:10.3390/jcm10235619_

Round 1

Reviewer 1 Report

The authors presented their findings that functional and/or anatomical differences between the right and left eyes may be involved in their results.

Their results of this study are very interesting.

Only reading at these results, many ophthalmologists will have doubts. But when the ophthalmologists read the discussion, a lot of ophthalmolgists find it very logical and convincing. 

From a clinical perspective, this paper will contribute to the knowledge base of various ocular circulatory disease.

Their is a mistake in the description of the reference. 

Reference 19, this paper was not published in 2010, it is 2020.

Author Response

November 13, 2021

Dear Reviewer #1

We submit our revised manuscript titled, “Background Factors Affecting Visual Acuity at Initial Visit in Eyes with Central Retinal Vein Occlusion: Multicenter Study in Japan” to be re-considered for publication in Journal of Clinical Medicine. We have revised our manuscript according to your suggestion. The revised parts are shown in red font in the revised manuscript.

Comment: There is a mistake in the description of the reference. Reference 19, this paper was not published in 2010, it is 2020.

Answer: Thank you. We have corrected this (see Line 455).

We look forward to hearing from you regarding our submission. We would be glad to respond to any further questions and comments that you may have. Thank you very much for your consideration.

Sincerely,

Mineo Kondo, MD, PhD, FARVO

Department of Ophthalmology, Mie University Graduate School of Medicine.

2-174 Edobashi, Tsu, Mie 514-8507, Japan

E-mail: mineo@clin.medic.mie-u.ac.jp

TEL: +81-59-231-5027  FAX: +81-59-231-3036

Reviewer 2 Report

This paper does not bring significant contribution to the knowledge in the field. Originality is low and there are shortcomings in the methodology. For instance, the CMT is measured with different devices in the centers involved in the study and therefore cannot be compared. The authors state that " the visual prognosis of CRVO is still not any better than that for branch retinal vein occlusion (BRVO)". The worse prognosis of CRVO  as compared to BRVO is not arguable. Also the definition as "ischemic" CRVO is made not uniformly and according to mixed criteria.

Author Response

November 13, 2021

Dear Reviewer #2

We submit our revised manuscript titled, “Background Factors Affecting Visual Acuity at Initial Visit in Eyes with Central Retinal Vein Occlusion: Multicenter Study in Japan” to be re-considered for publication in Journal of Clinical Medicine. We have revised our manuscript according to your suggestions. The revised parts are shown in red font in the revised manuscript.

Comment 1: The CMT is measured with different devices in the centers involved in the study and therefore cannot be compared.

Answer: Thank you for your valuable comments. In this study, we collected clinical data of 517 CRVO patients from 17 different ophthalmological institutions retrospectively. Therefore, it was difficult to limit to a single type of OCT device. According to your suggestion, we have investigated what kind of OCT device was used in all 17 institutes. We found that four OCT devices were used in our 17 institutes. A chi-square test was performed to analyze whether a particular OCT device was used more frequently for the right or left eye. We found that there was no significant trend that a particular OCT device was used more frequently for the right or left eye (P=0.664). Based on these results, we think that it is unlikely that the differences in the OCT device used caused the left-right differences in the CMT. We have added these results in the Discussion (Line 349-355). We thank you for this suggestion.

Comment 2: The authors state that "the visual prognosis of CRVO is still not any better than that for branch retinal vein occlusion (BRVO)". The worse prognosis of CRVO as compared to BRVO is not arguable.

Answer: We agree and have replaced this sentence as “the visual prognosis of CRVO is still not good” (Line 64).

Comment 3: The definition as "ischemic" CRVO is made not uniformly and according to mixed criteria.

Answer: We did not use a uniform definition for the ischemic-type of CRVO, but used a modified definition using the combination of various clinical findings. This is because this was a retrospective study using medical records. Initially, we tried to define the ischemic-type CRVO based on the results of the fluorescein angiography used in the CVO Study (1995). However, our preliminary results showed that fluorescein angiography was not performed in more than one-half of our CRVO cohort. Therefore, for eyes with CRVO where fluorescein angiography was not performed, we tried to determine the ischemic-type comprehensively by combining the results of several clinical findings based on a previous paper by Hayreh et al. (1990). They demonstrated that the combination of several clinical findings can be used to determine the ischemic-type CRVO with a high degree of accuracy. Actually, Brown et al. used this definition as the inclusion criterion for ischemic-type CRVO in the RAVE Study (2014). Because this study was a retrospective study and we had to determine the ischemic-type based on the medical records, we had to use these methods. Your comments have been added as a major limitation of this study in the Discussion section (Line 356-366).

We look forward to hearing from you regarding our submission. We would be glad to respond to any further questions and comments that you may have. Thank you very much for your consideration.

Sincerely,

Mineo Kondo, MD, PhD, FARVO

Department of Ophthalmology, Mie University Graduate School of Medicine.

2-174 Edobashi, Tsu, Mie 514-8507, Japan

E-mail: mineo@clin.medic.mie-u.ac.jp

TEL: +81-59-231-5027      FAX: +81-59-231-3036

Reviewer 3 Report

Mineo Kondo et al. conducted an interesting study on 517 eyes with CRVO collected over 17 different centers. They found that poorer initial BCVA was associated older age and right eye-involvement. It may be not difficult to understand why oder age was asscociated with a poorer initial VA, since older age itself is generally associated with poorer vision, even in healthy subjects. Furthermore, you would expect more sclerotic and degenerative changes in aged vessels as discussed in the manuscript. However, right eye-involvment, though it may be a novel, interesting finding, needs more discussion. 

First, the right eye-CRVO had thicker central retina (i.e. greater macular edema) than the left eye, which appears to the be reason for vision difference. It then, the right question we need to ask is whether right eye is involved with more "severe" type of CRVO that acommpanies thicker central retina. This does not necesarrily mean that ischemic CRVO would be more often in the eye with greater CMT, as ischemic CRVO can present with lesser CMT, but still shows greatly reduced visual acuity. Since there was no difference in the rate of ischemic CRVO between the eyes, vision difference is not attributed to "type" of CRVO, but to "degree" of CMT. Since patients  with less than 12 months of the interval between the symptom onset and  the initial visit were included in the study, there could be a wide arrange of the interval time. Because  the macular edema, especially in non-ischemic CRVO, can somtimes improve over time, authors need to see if there was any difference in this interval periods between the eyes and incorporate this factor into the multivariate analysis. 

Second, it was stated in the method that the eye with an earlier onset was included in bilateral cases. For the sake of statistical methods, only one eye needs to be included in a subject. However, to prove that occurrence of CRVO in the right eye presents with poorer vision, these bilateral cases would be more desirable to study. It would be interesting to conduct a sub-group analysis to see if right eye vision tends to be poorer in bilateral cases. If there are not many bilateral CRVO cases for statistical analysis, it would be interesting to see if right eye vision is still poorer after including both eyes for the analysis of right eye-left eye hypothesis.   

Third, left common carotid artery (CCA) stems directly from the arch of aorta and is more affected by aortic arch pressure (hydrostatic pressure), whereas the right CCA stems from the  extension (innominate artery) of the ascending aorta is subjected to pressure from the aascending aortic blood flow (dynamic pressure). It would be more interesting to see if authors findings could be explained in terms of anatomical changes of left and right CCA .

Author Response

November 13, 2021

Dear Reviewer #3,

We submit our revised manuscript titled, “Background Factors Affecting Visual Acuity at Initial Visit in Eyes with Central Retinal Vein Occlusion: Multicenter Study in Japan” to be re-considered for publication in Journal of Clinical Medicine. We have revised our manuscript according to your suggestions. The revised parts are shown in red font in the revised manuscript.

Comment 1: The right eye-CRVO had thicker central retina (i.e. greater macular edema) than the left eye, which appears to the be reason for vision difference. It then, the right question we need to ask is whether right eye is involved with more "severe" type of CRVO that accompanies thicker central retina. This does not necessarily mean that ischemic CRVO would be more often in the eye with greater CMT, as ischemic CRVO can present with lesser CMT, but still shows greatly reduced visual acuity. Since there was no difference in the rate of ischemic CRVO between the eyes, vision difference is not attributed to "type" of CRVO, but to "degree" of CMT. Since patients with less than 12 months of the interval between the symptom onset and the initial visit were included in the study, there could be a wide arrange of the interval time. Because the macular edema, especially in non-ischemic CRVO, can sometimes improve over time, authors need to see if there was any difference in this interval periods between the eyes and incorporate this factor into the multivariate analysis. 

Answer: Thank you for your suggestions. We agree and compared the interval from symptom onset to initial visit to the hospital between the right and left eye. Mean interval from symptom onset to initial visit to the hospital was slightly longer in the right eye (6.8 ± 7.0 weeks) than in the left eye (6.1 ± 6.9 weeks), but this difference was not statistically significant (P=0.251). We added these results in Line 268-272. We also confirmed that the interval from symptom onset to initial visit to the hospital was not significantly associated with the visual acuity at the initial visit in the multivariate regression analysis (β = -0.008, P = 0.849, see Table 2).

Comment 2: It was stated in the method that the eye with an earlier onset was included in bilateral cases. For the sake of statistical methods, only one eye needs to be included in a subject. However, to prove that occurrence of CRVO in the right eye presents with poorer vision, these bilateral cases would be more desirable to study. It would be interesting to conduct a sub-group analysis to see if right eye vision tends to be poorer in bilateral cases. If there are not many bilateral CRVO cases for statistical analysis, it would be interesting to see if right eye vision is still poorer after including both eyes for the analysis of right eye-left eye hypothesis.

Answer: Thank you for your suggestions. We reviewed all of the data again and found that we had 11 patients who had CRVO in both eyes during our observational period of 12 months. Because the number of sample (n = 11) was too small to compare the visual acuity between the right and left eye in this subgroup, we compared the visual acuity of right and left eyes in all CRVO eyes including the data of both eyes of these 11 patients. We confirmed that the difference in visual acuity between the right and left eyes was still statistically significant (P=0.020). We have added these results in Line 243-246.

Comment 3: Left common carotid artery (CCA) stems directly from the arch of aorta and is more affected by aortic arch pressure (hydrostatic pressure), whereas the right CCA stems from the extension (innominate artery) of the ascending aorta is subjected to pressure from the ascending aortic blood flow (dynamic pressure). It would be more interesting to see if authors findings could be explained in terms of anatomical changes of left and right CCA.

Answer: Thank you. We believe that the anatomical differences of the right and left carotid artery you suggested could have been one of the factors which caused the left-right difference in the visual acuity of CRVO. We have added your comments in our Discussion section (Line 330-337).

We look forward to hearing from you regarding our submission. We would be glad to respond to any further questions and comments that you may have. Thank you very much for your consideration.

Sincerely,

Mineo Kondo, MD, PhD, FARVO

Department of Ophthalmology, Mie University Graduate School of Medicine.

2-174 Edobashi, Tsu, Mie 514-8507, Japan

E-mail: mineo@clin.medic.mie-u.ac.jp

TEL: +81-59-231-5027   FAX: +81-59-231-3036

Round 2

Reviewer 2 Report

Dear Authors,

I fell that your study does not meet the criteria to be published in Journal of Clinical Medicine and I therefore suggest you to send it for being considered in another medical journal. It does not bring significant contribution to the knowledge in the field and it does not provide any original perspective on the subject.

Reviewer 3 Report

Please state that whether right eye's or left eye's visual acuity was lower (line 245-246) for the sake of clarity, though it is assumed that right eye's vision is lower.